# Auricular Vagus Nerve Stimulation Improves Visceral Hypersensitivity and Gastric Motility and Depression-like Behaviors via Vago-Vagal Pathway in a Rat Model of Functional Dyspepsia

**DOI:** 10.3390/brainsci13020253

**Published:** 2023-02-01

**Authors:** Liwei Hou, Peijing Rong, Yang Yang, Jiliang Fang, Junying Wang, Yu Wang, Jinling Zhang, Shuai Zhang, Zixuan Zhang, Jiande D. Z. Chen, Wei Wei

**Affiliations:** 1Institute of Acupuncture and Moxibustion, China Academy of Chinese Medical Sciences, Beijing 100700, China; 2Wangjing Hospital, China Academy of Chinese Medical Sciences, Beijing 100102, China; 3Guang’anmen Hospital, China Academy of Chinese Medical Sciences, Beijing 100053, China; 4Division of Gastroenterology and Hepatology, University of Michigan, Ann Arbor, MI 48109, USA

**Keywords:** vagotomy, vagus nerve, cholinergic anti-inflammatory pathway, inflammation, hypothalamic–pituitary-adrenal axis

## Abstract

Transcutaneous auricular vagus nerve stimulation was recently reported to have a therapeutic potential for functional dyspepsia (FD). This study aimed to explore the integrative effects and mechanisms of auricular vagus nerve stimulation (aVNS) in a rodent model of FD. Methods: We evaluated the effects of aVNS on visceral hypersensitivity, gastric motility and open field test (OFT) activity in iodoacetamide (IA)-treated rats. The autonomic function was assessed; blood samples and tissues were collected and analyzed by an enzyme-linked immunosorbent assay and western blot. Vagotomy was performed to investigate the role of vagal efferent nerve. Results: aVNS reduced the electromyography response to gastric distension, improved gastric emptying and increased the horizontal and vertical motion scores of the OFT in IA-treated rats. The sympathovagal ratio was increased in IA-treated rats but normalized with aVNS. The serum cytokines TNF-α, IL-6, IL-1β and NF-κBp65 were increased in IA-treated rats and decreased with aVNS. The hypothalamus–pituitary–adrenal axis was hyperactive in IA-treated rats but inhibited by aVNS. The expression of duodenal desmoglein 2 and occludin were all decreased in IA-treated rats and increased with aVNS but not sham-aVNS. Vagotomy abolished the ameliorating effects of aVNS on gastric emptying, horizontal motions, serum TNF-α and duodenal NF-κBp65. Conclusion: aVNS improves gastric motility and gastric hypersensitivity probably by anti-inflammatory mechanisms via the vago-vagal pathways. A better understanding on the mechanisms of action involved with aVNS would lead to the optimization of the taVNS methodology and promote taVNS as a non-pharmacological alternative therapy for FD.

## 1. Introduction

Functional dyspepsia (FD) is a debilitating functional gastrointestinal (GI) disorder characterized by early satiety, post-prandial fullness or epigastric pain related to meals without organic etiology, and is present for at least 6 months [1]. The disease affects 8% to 12% of the general population [2]. FD was perceived to benefit most from pharmacotherapy although most treatments have modest effects [1]. An adverse event profile of pharmacotherapy has recently been recognized in the guideline [3]. A proton pump inhibitor (PPI) is recommended with high quality evidence as the empirical therapy. Nevertheless, gastric acid is not the main risk factor, and up to 50% of patients with postprandial distress syndrome (PDS) seek other treatment [4]. Prokinetic therapy is suggested if dyspepsia patients are not responding to PPI, but with conditional recommendation and low-quality evidence. In addition, some prokinetics have significant risks of adverse events [3]. Due to a lack of efficacy and/or side effects, FD patients treated with medical therapies often continue to consult and desire further treatment. The economic impact is estimated to be over USD 18 billion per year in the USA [5]. Accordingly, further exploration of the other forms of safe and effective therapies for FD is crucial.

The pathophysiology of FD is multifactorial [6,7]. The symptoms associated with PDS were thought to originate from gastric motor dysfunction, and the symptoms associated with epigastric pain syndrome (EPS) were thought to be due to mechanical hypersensitivity of the stomach [1]. In addition, chemical hypersensitivity related to exogenous or endogenous acid in the duodenum and the decreased clearance of acid has been associated with nausea. Neither PPI nor prokinetic treatment has been proven effective for reducing visceral hypersensitivity. Low-dose tricyclic antidepressants were suggested as a therapy because of their peripheral pain-modifying effect but are accompanied with adverse events such as constipation, dry mouth, urinary retention and/or somnolence [8]. At present, there is no medication that is effective for improving both gastric dysmotility and visceral hypersensitivity.

Auricular vagus nerve stimulation (aVNS), or transcutaneous aVNS (taVNS), is a neuromodulation method that involves the application of electrical currents through surface electrodes targeting the auricular branch of the vagus nerve (ABVN). A few clinical studies have shown the ameliorating effects of taVNS on dyspeptic symptoms with a good safety profile [9,10]. A preclinical study of auricular electroacupuncture consisted of sterile acupuncture needles inserted into the bilateral auricular acupoints, which are located between the cymba conchae and cavum conchae ameliorated gastric hypersensitivity in a rodent model of FD, and the effect was attributed to the improvement of sympathovagal balance [11]. Nevertheless, it is unknown whether aVNS or taVNS is capable of improving both gastric dysmotility and visceral hypersensitivity associated with FD and, furthermore, mechanisms involved in the ameliorating effects of aVNS/taVNS remain largely unclear.

Ford AC et al. [1,12,13,14] proposed one disease model for the pathophysiology of FD: (1) central signaling, including via corticotropin-releasing hormone and neurotransmitters, can alter peripheral gastrointestinal function. (2) Alterations in the balance between environmental factors and the gastrointestinal tract lumen and gut microbiome state might lead to dysfunction of the intestinal epithelial barrier, increasing mucosal integrity. Antigens might then be recognized by immune cells, leading to a low-grade inflammatory response. Inflammatory mediators and cytokines released by activated intestinal eosinophils and mast cells might sensitize enteric nerves, thereby causing visceral hypersensitivity and motor dysfunction.

From the above, we hypothesize that aVNS improves FD via regulating central signaling and low-grade inflammatory mediated by the vago-vagal pathway. In this study, we used a previously validated rat model of FD [15] to explore the effects of aVNS on gastric motility and visceral pain, as well as depression-like behaviors and mechanisms involving anti-inflammatory pathway and the hypothalamic–pituitary–adrenal (HPA) axis.

## 2. Materials and Methods

### 2.1. Animals

Five-day-old, male Sprague Dawley (SD) rats (Sibeifu, Beijing, China) were purchased and housed in animal cages (35 cm × 25 cm × 20 cm; before ablactation: a litter/cage; after ablactation (three-week-old) until to adult: 3 rats/cages; after surgery: a single rat/cage) under the following conditions: room temperature of 23 °C ± 2 °C, relative humidity of 60% ± 5% and 12 h light/dark cycles with an automatic air circulation system. The rats had free access to conventional rodent lab chow and water. Experiments were initiated when the rats were 8 weeks old.

### 2.2. Model of “FD”

Rats in various “FD” groups were treated as follows: ten-day-old SD rats received 0.2 mL 0.1% iodoacetamide (IA) in 2% sucrose daily for 6 days and housed normally to grow until adulthood of 8 weeks old [15]. Rats in the control group did not receive any treatment.

### 2.3. Experimental Protocols

The experiment aimed to evaluate effects and mechanisms of aVNS in the IA-treated rats and performed in 4 groups of rats (36 rats):Control group: normal rats without any IA treatment and aVNS or sham, but received balloon and wires implantation of EMG and ECG (*n* = 12; 6 rats were randomly picked and sacrificed for the gastric emptying test).IA-treated group: IA-treated rats received balloon and wires implantation of EMG and ECG but not aVNS or sham (*n* = 12; 6 rats were randomly chosen and sacrificed for the gastric emptying test).aVNS group: IA-treated rats received aVNS and balloon and wires implantation of EMG and ECG (6 IA-treated rats with daily aVNS).Sham-aVNS group: IA-treated rats received sham-aVNS and balloon and wires implantation of EMG and ECG (6 IA-treated rats with daily sham-aVNS).

The experimental protocol for these animals is shown in Figure 1A (not every animal received all indicated treatments or procedures). Surgical procedures were performed at the age of 8 weeks for placement of gastric balloon and various wire electrodes. Physiological measurements, including gastric emptying and open field test (OFT), were made before (9th week) and after (11th week) the treatment (aVNS or sham-aVNS), while electromyography (EMG) was performed at the 9th week and 10th week. Various tissues were collected at the end of the study for mechanistic assessment.

In order to investigate the vagal efferent pathway involved in the ameliorating effects of aVNS in IA-treated rats, we implemented another experiment including 4 groups (24 rats): (1) the IA-treated group and IA + aVNS group were similar with above experiment (*n* = 6); (2) IA + Subphrenic vagotomy (SV) group (6 IA-treated rats were subjected to SV without stimulation) and (3) IA + aVNS + SV group (6 IA-treated rats were subjected to SV with aVNS). The experimental protocol for these rats was similar to the other four groups of rats (Figure 1B) except that SV was performed but no gastric balloon or EMG electrodes were implanted (this was to avoid excessive surgeries which would lead to severe complications).

### 2.4. Surgical Procedures

At the age of 8 weeks, the rat, after overnight fasting, was operated under anesthesia by inhalation of 2.5% isoflurane for the following procedures:

Electrode implantation for aVNS or sham-aVNS: the left ear or right ear of the rat was disinfected with 75% alcohol. One pair of wire electrodes (Medtronic, Minneapolis, MN, USA) was fixed into the cymba conchae and cavum conchae.

Distension balloon implantation and electrodes implantation for EMG and ECG recording: the methods are available in previous publications [11,16,17]. See Appendix A for further details.

SV: an incision along the mid-abdominal line was made. Bilateral subphrenic vagus nerves were identified under a microscope (OLYMPUS, Tokyo, Japan) and severed.

The intramuscular injection of butorphanol tartrate (0.03 mg/kg, once, 5 min before surgery, UK Bless Pet Animal Protection Limited, South Croydon, UK) was given to relieve postoperative pain. After the surgery, the rat was single-caged and given one week to recover before the initiation of any experiment.

### 2.5. Auricular Vagus Nerve Stimulation

Electrical stimulation was performed by a watch-size stimulator (Ningbo Maida Medical Device, Inc., Ningbo, China). The positive and negative poles of the stimulator were connected to the wires attached to the auricular concha. During stimulation, the rat was kept in a restrainer in which the rat could only move its legs slightly and could not turn around (Figure 1C). aVNS was delivered using the parameters of 0.5 mA, 0.1 s-on and 0.4 s-off and 100 Hz, and 30 min daily for 2 weeks (9:00 am~10:00 am). This parameter setting was previously shown to improve visceral hypersensitivity in rats [11]. Sham-aVNS was performed under the same experimental setting except that the stimulation output was set at 0 mA.

### 2.6. Measurements

The methods for the EMG, ECG, gastric emptying and OFT are available in previous publications [11,16,17]. See Appendix A for further details.

#### 2.6.1. Assessment of Serum Cytokines Tumor Necrosis Factor α (TNF-α), Interleukin 6 (IL-6), Interleukin 1β (IL-1β), Adreno-Cortico-Tropic-Hormone (ACTH) and Corticosterone

At the end of the study, 90 min after feeding for assessing gastric emptying, all rats were anesthetized by general 2.5% isoflurane. Blood samples were collected from the abdominal aorta, centrifuged for 10 min (high-speed refrigerated centrifuge, Eppendorf, Hamburg, Germany) at 3000 r/min, and the serum was reserved for test. The serum level of TNF-α (BLUE GENE, E02T0008, Shanghai, China), IL-6 (BLUE GENE, E02I0006, Shanghai, China), IL-1β (BLUE GENE, E02I0010, Shanghai, China), ACTH (raybiotech, Atalanta, GA, USA) and corticosterone (abcam, Waltham, MA, USA) were assessed by corresponding ELISA kits according to the manufacturer’s protocol.

#### 2.6.2. Gastric Tissue Acetylcholine (Ach)

The gastric antrum tissue was collected. The tissue was ground with the RIPA Buffer and protease inhibitor. After being centrifuged, the supernatant was taken for the assessment of total protein. Tissue Ach was tested by an Ach Elisa kit (BLUE GENE, Shanghai, China) according to the manufacturer’s protocol.

#### 2.6.3. Western Blotting

At the end of the study, the gastric antrum tissue, duodenal tissue, hypothalamus and hippocampus tissue in all the rats were collected. A western blot was used to detect the quantitative expressions of achreceptor M_3_ muscarinic receptor (M3R), hypothalamus corticotropin releasing factor (CRF), amygdaloid nucleus corticotropin releasing factor one (CRF1) and duodenal nuclear factor kappa Bp65 (NF-κBp65). See Appendix A for further details.

### 2.7. Statistical Analysis

SPSS 25.0 software was used for statistical analysis. Measurement data are expressed in the form of (Mean ± SD). In case the data were consistent with normality and homogeneity variance, the independent sample t-test was used for comparison between two groups. A paired t-test was adopted for comparison within the same sample before and after the intervention. One-way ANOVA was used for comparison among multiple groups, and the Fisher’s least significant difference (LSD) method was used for post hoc test. Tamhane’s T2(M) method was used after ANOVA if the data were not in accordance with homogeneity of variance. If the data did not conform to normality, the non-parametric test was adopted. Statistical analysis figure was performed using GraphPad Prism software (version 7). The level of test was α = 0.05, and *p* < 0.05 was considered as statistically significant.

## 3. Results

### 3.1. Neonatal IA Treatment-Induced Visceral Hypersensitivity, Gastric Dysmotility and Decreased OFT Scores

As the pressure increases, the EMG response to gastric distension (GD) becomes more obvious (Figure 2a), especially in IA-treated rats. Compared with control rats, the EMG change ratio was significantly increased in IA-treated rats from 20 mmHg to 80 mmHg (*p* = 0.04 for 20 mmHg, *p* = 0.02 for 40 mmHg, *p* = 0.001 for 60 mmHg and *p* = 0.04 for 80 mmHg) (Figure 2b). In IA-treated rats, no difference in the EMG responses to GD was noted at 20 mmHg and 40 mmHg among three groups (IA, sham-aVNS and aVNS) at baseline before the two-week treatment (ninth week in Figure 1A and Figure 2b). The IA-treated rats also showed a lower gastric emptying rate than control rats (42.3 ± 4.4 vs. 63.7 ± 4.5, *p <* 0.0001) (Figure 2d). The results demonstrated gastric hypersensitivity and dysmotility in IA-treated rats.

The OFT was used to quantitatively evaluate the loco-motor activity that is known to be associated with animal behaviors of anxiety and depression. Both horizontal and vertical scores were decreased in the IA-treated rats compared with the control rats (both *p* < 0.0001) (Figure 2d). There were also no significant differences in rats between the IA-treated group, the aVNS group and the sham-aVNS group. The results showed depression-like behavior in IA-induced rats.

### 3.2. Effects of aVNS on Gastric Hypersensitivity, Gastric Dysmotility and Depression-Like Behaviors

At the end of the one-week treatment, the EMG response to GD in the aVNS group was significantly lower than that in the IA-treated rats without aVNS or sham-aVNS at 60 mmHg (*p* = 0.003, *p* = 0.04) and 80 mmHg (*p* = 0.04, *p* = 0.04) in Figure 3A. It was also noted that at the end of the one-week treatment, no difference was noted in the EMG response to GD at any pressure between the aVNS group and the control rats, suggesting a normalization of IA-treatment-induced visceral hypersensitivity. Longitudinally (a comparison between the baseline and after the treatment within the same group), the one-week aVNS reduced the EMG response to GD at 40 mmHg (*p* = 0.005), 60 mmHg (*p* = 0.003) and 80 mmHg (*p* = 0.006) in Figure 3B; the one-week sham-aVNS resulted in no significant changes in the EMG response to GD at any pressure in Figure 3C. These results demonstrated the ameliorating effect of aVNS on gastric hypersensitivity.

The aVNS also improved gastric emptying in comparison with both the IA group (*p* = 0.04) and the sham-aVNS group (*p* < 0.001) (Figure 3D).

aVNS increased vertical scores and the horizontal scores but not sham-aVNS (*p* < 0.0001, *p* < 0.0001 both compared with control rats; *p* = 0.004, *p* < 0.0001 compared with sham-aVNS rats) (Figure 3E,F). Longitudinally (a comparison between the baseline and after the treatment within the same group), the two-week aVNS increased the vertical scores (*p* = 0.01) (Figure 3G); however, the two-week sham-aVNS resulted in no increasing changes in the horizontal scores and vertical scores in Figure 3H. The aVNS relieved the depression-like or anxiety-like behavior of model rats but not the sham-aVNS.

### 3.3. aVNS Improved Vagal Activity and Gastric Tissue Level of Ach and Expression of Its Receptor

The LF/HF (on behalf of sympathovagal ratio) from ECG was increased and the HF/(HF + LF) (on behalf of vagal activity) was reduced in IA-treated rats (*p* = 0.001, *p* = 0.03. both vs. control rats). These abnormalities were normalized with aVNS rats (*p <* 0.001, *p* = 0.001. both vs. IA-treated rats). The sham-aVNS could not regulate the disorder (Figure 4A,B). The ECG wave was shown in Figure 4C. We detected gastric tissue levels of Ach, and expression of its receptor M3R in gastric tissue for combination of acetylcholine and its receptor can promote gastric motility. Release of Ach and expressions of its receptor M3R In gastric tissue were both reduced in IA-treated rats (*p* = 0.002, *p* = 0.003. both vs. control rats) and improved with aVNS (*p* = 0.001, *p* = 0.023 both vs. IA-treated rats; *p <* 0.001, *p* = 0.035 vs. sham-aVNS rats) but not with sham-aVNS (Figure 4D,E).

### 3.4. aVNS Suppressed Inflammation and Improved Impaired Mucosal Integrity

The expression of duodenal NF-κBp65 was increased in IA-treated rats (0.87 ± 0.26, *p* < 0.01 compared with control rats) and decreased with aVNS (0.34 ± 0.17, *p* < 0.01) but not with the sham-aVNS (Figure 5A). The serum levels of inflammatory cytokines TNF-α, IL-1β and IL-6 were increased in model rats (*p* < 0.0001, *p* < 0.0001, *p* < 0.01, compared with control rats for all) and decreased with aVNS (*p* < 0.0001, *p* < 0.0001, *p* < 0.01, compared with IA-treated rats for all). There were no differences between sham-aVNS rats and IA-treated rats (Figure 5B–D). The IA-treated rat model was in low-grade inflammation. aVNS alleviated the low-grade inflammation, probably via activating the anti-inflammation of the vagus nerve.

The expression of duodenal desmoglein 2 (DSG2), β-catenin, and tight junction ligandins occludin were all decreased in IA-treated rats (*p* = 0.005, *p* = 0.02, *p* = 0.02 all compared with control rats). The two-week aVNS increased the expression of DSG2 and occludin proteins (*p* = 0.03, *p* = 0.003 both compared with IA-treated rats) (Figure 5E,F). Though there were no differences in the expression of β-catenin among IA-treated, aVNS and sham-aVNS groups, the aVNS had an increasing tendency (Figure 5G). In addition, aVNS seemed to enhance the expression of zonula occluden (ZO-1) (Figure 5H). These results demonstrated the ameliorating effect of aVNS on mucosal integrity.

### 3.5. aVNS Inhibited Hyperactivation of the HPA Axis

The HPA axis is an endocrine system activated in response to stress, as well as an important part of the gut-brain axis. During stress factor exposure, CRF is released from the hypothalamus and promotes the secretion of ACTH from the pituitary gland to stimulate the release of glucocorticoids (cortisol in humans or corticosterone in rodents) from the adrenal cortex [18].

The CRF in the hypothalamus was higher in IA-treated rats (*p* = 0.017. vs. control rats) but reduced with aVNS (*p* = 0.012) (Figure 6A). The CRF1 in the amygdaloid nucleus was higher in IA-treated rats (*p* < 0.001. vs. control rats) but reduced with aVNS (*p* = 0.01) (Figure 6B). ACTH and corticosterone in serum were both increased in IA-treated rats (*p* < 0.001, *p* < 0.001. both vs. control rats) and reduced with aVNS but not sham-aVNS rats) (*p* = 0.001, *p* < 0.001. both vs. IA rats; *p* = 0.015, *p* = 0.025 both vs. sham-aVNS rats). (Figure 6C,D). The results showed hyperactivation of the HPA axis in IA-treated rats (ACTH and corticosterone but not CFR or CRF1). aVNS inhibited the hyperactivation of the HPA axis significantly. We correlated the HPA measures (ACTH, corticosterone) to the OFT behaviors for each rat (Figure 6E–H). The results showed the negative linear correlation between HPA measures and OFT behaviors.

### 3.6. Vagotomy Abolished the Ameliorating Effect of aVNS on Gastric Emptying, and Horizontal Motions

To test the roles of vagus nerve, the subphrenic vagus nerves on both sides of the esophagus in IA-treated rats were cut off, which specifically innervates gastric motility, then we observed the effects of the aVNS. The gastric emptying in IA + SV rats was lower than IA-treated rats, but there were no significant differences between the two groups. From Figure 7A, we can see that aVNS improved gastric emptying in IA-treated rats (*p* = 0.007, IA + aVNS vs. IA). This improvement was, however, not noted in the rats with SV (*p* > 0.05, IA + aVNS + SV vs. IA + SV), suggesting the vagally mediated effect of aVNS.

aVNS improved horizontal score (*p* = 0.004 compared with IA-treated rats), but the horizontal scores decreased in IA + aVNS + SV rats. Vertical scores had similar trends although there were no significant differences among four groups (*p* = 0.027 compared with the aVNS rats) (Figure 7B,C). The SV abolished the ameliorating effects of aVNS on gastric dysmotility and horizontal motions.

### 3.7. Vagotomy Abolished the Anti-Inflammation of aVNS

We further investigated the regulation of aVNS on the anti-inflammation in IA-treated rats with SV. Compared with aVNS rats, the duodenal NF-κBp65 increased in IA-treated rats with aVNS received SV simultaneously (*p* < 0.05, vs. aVNS group) (Figure 7D). The release of Ach was increased with aVNS (*p* = 0.013 compared with IA-treated rats) but reduced in IA + aVNS + SV rats (*p* = 0.001 compared with aVNS rats) (Figure 7E). There was no significant difference of serum TNF-α among IA-treated rats, IA + SV rats and IA + aVNS + SV rats. The aVNS reduced the release of TNF-α in serum (*p* < 0.0001 compared with IA-treated rats) but aVNS + SV did not (*p* < 0.0001 compared with aVNS rats) (Figure 7F). The vagotomy blocked the anti-inflammation of the vagus nerve.

## 4. Discussion

### 4.1. The Iodoacetamide Induced FD-Like Behaviors in Rats

In this study, we used the validated IA-induced rat model of FD to assess the integrative effects and autoimmune mechanisms of aVNS for FD. Early-life adverse events ranging from physical, chemical, psychological to environmental stressors are linked to visceral hypersensitivity in later life [19]. IA, a protease inhibitor, was used to induce a transient mild gastric inflammation in neonatal rats [15,20]. The neonatal GI function is vulnerable, and stimulation by IA can lead to chronic visceral hypersensitivity and a long-lasting increase in depression- and anxiety-like behaviors that persist into adulthood [21]. Consistent with previously published data, our findings also demonstrated visceral hypersensitivity, gastric dysmotility and depression-like behaviors in this rodent model of FD.

### 4.2. The Integrative Effects of aVNS for FD-Like Rats

aVNS improved or normalized IA-induced visceral hypersensitivity, delayed gastric emptying and animal behaviors suggested depression. These integrative effects suggested a unique therapeutic potential of the proposed method for FD. The pathophysiology of FD is complex, involving gastrointestinal sensory and motor dysfunction, immune dysfunction and gut–brain axis dysfunction [1]. The findings of the present study indicated that the aVNS method was effective in improving all these pathophysiological factors. It should be noted that although there are various therapies for FD [3,8,22], there is a lack of therapies that exert integrative or comprehensive effects on various pathophysiologies of FD. For example, prokinetic agents may improve gastric emptying but do not reduce visceral hypersensitivity [22], whereas neuromodulators may be used to ameliorating visceral hypersensitivity but do not improve gastric motility [8].

### 4.3. The Ameliorating Effect of aVNS on Visceral Hypersensitivity Probably by Activating the Anti-Inflammation and Improving the Mucosal Integrity in Duodenum

Visceral hypersensitivity is known to lead to visceral pain. However, there is a lack of safe and effective pharmacological therapy to suppress visceral hypersensitivity. In a previous study, Zhou J et al. found that aVNS with certain parameters significantly reduced gastric hypersensitivity in the same rodent model of FD, and the effect was attributed to the improvement of sympathovagal balance [11]. In one study, transcutaneous cervical vagal nerve stimulation was reported to show improvement in pain symptom subscales in patients with gastroparesis, and the effect was hypothesized to be associated with the modulation of reflex parasympathetic activity [10]. In another clinical study, the modulation of vagal tone was reported to enhance gastroduodenal motility and reduce somatic pain sensitivity [9]. Due to its extensive innervation of the stomach and its predominant role in parasympathetic regulation of inflammation and motility, the vagus nerve is considered as a powerful therapeutic target to reduce visceral hypersensitivity [23]. In the present study, aVNS enhanced vagal efferent activity and released Ach, demonstrating the involvement of the vagus nerve in the amelioration of aVNS for FD. Anti-inflammation is one important anti-inflammatory function of the vagus nerve and a neural pathway advanced by the identification of a neural mechanism that inhibits macrophage activation through vagus nerve efferents [24]. Efferent activity in the vagus nerve leads to Ach release in organs of the reticuloendothelial system, including the liver, heart, spleen and gastrointestinal tract [24]. Ach interacts with a-bungarotoxin-sensitive nicotinic receptors on tissue macrophages, which inhibit the NF-κBp65 signal inflammation pathway, then reduce the release of TNF, IL-1 and other cytokines [24,25]. In our study, the NF-κBp65 in the duodenal was increased in the IA-treated rats. This low-grade inflammation was inhibited by aVNS. Furthermore, these effects were abolished by vagotomy. All these findings demonstrated the anti-inflammation of aVNS was mediated probably via the vago-vagal reflex. In addition to this, the intestinal epithelial barrier was damaged, reflected as decreased intestinal tight junction and adhesive proteins in the IA-treated rats, but enhanced with aVNS. In brief, aVNS reduced visceral hypersensitivity, probably via activating anti-inflammation of the vagus nerve and increasing mucosal integrity, leading to the relieving of visceral pain.

### 4.4. aVNS Promoted the Gastric Motility Probably by Improving Vagal Activity

In our study, aVNS improved gastric motility probably by activating the vagus nerve. Similarly, in a clinical study, Zhu Y et al. reported symptom relief in patients of FD with taVNS by improving gastric dysmotility and vagal activity [10]. Our findings were consistent with the clinical study. Anatomically, auricle areas are innervated by the ABVN in varying degrees [26]. The ABVN is known to project to the nucleus tractus solitarius (NTS) [27]. Accordingly, aVNS may activate neurons in the NTS that projects to other nuclei of the brain. The dorsal motor nucleus of the vagus nerve (DMV), on the other hand, receives signals from other parts of the brain and transmits vagal efferent signals down to the stomach and other peripheral organs. In IA-treated rats, the vagal nerve activity was reduced, which could lead to delayed gastric emptying. aVNS improved gastric emptying by enhancing vagal efferent activity, probably. The abolishment of the effect of aVNS by vagotomy further demonstrated that the integrity of the vagus nerve was important.

### 4.5. aVNS Alleviated Depression-Like Behaviors Probably via Downregulating the Hyperactivity HPA Axis or Brain CRF Signaling Pathway

The IA-treated rats in the present study exhibited a significant decrease in the horizontal and vertical scores in the OFT, suggesting depression-like behaviors [28,29,30], which was reversed with aVNS, probably by inhibiting hyperactivation of the HPA axis. Previously, taVNS was reported as a safe and cost-effective therapeutic method for depressive disorder [31]. Clinically, in an 11-year historic cohort study with 40,394 people, the incidence of FD was found to be 1.7-fold greater in the depressive cohort than that in the non-depression cohort [32]. A 12-year prospective longitudinal study reported that depression without GI symptoms at baseline was predictive of FD [33]. In a rodent study, aVNS was reported to significantly reverse depression-like behaviors by downregulating the hyperactivity of the HPA axis [34]. In our study, the similar beneficial effect of aVNS on downregulating the hyperactivity of the HPA axis was observed. It is considered that vagus nerve activity can be relayed to the medullary reticular formation, to the locus coeruleus and the hypothalamus.

The vagotomy abolished the ameliorating effects of aVNS on horizontal motions but not vertical scores, which hinted that vagus nerve may be one of the pathways in the ameliorating effect of aVNS for FD with depression-like behaviors except for the HPA axis. We speculated that depression-like behaviors in this model were not only caused by a central mechanism, but possibly also visceral pain. Brain CRF signaling plays a key role in stress-related alterations of gastrointestinal functions, including the inhibition of gastric acid secretion and gastric-small intestinal transit and visceral hypersensitivity, which are probably mediated through the autonomic nervous system [35,36,37,38,39]. It is possible that the vagotomy not only abolished the effect of aVNS for alleviating visceral pain, but also depression-like behaviors caused by visceral pain.

In our study, sham-aVNS was not effective in results except for the vertical scores, CRF1 and CRF expressions, which are related with the HPA axis. As we know, the HPA axis is an important bidirectional endocrine system activated in response to irritability. CRF and CRF1 were decreased in both groups of aVNS and sham-aVNS, suggesting that the increase (compared with the control rats) was probably attributed to stress effects of surgical and stimulation procedures.

### 4.6. Clinical Perspective

FD involves multiple pathophysiologies and there is no therapy that could treat all of these. Pathophysiologies of FD include hypersensitivity, gastrointestinal motor dysfunction, low-grade inflammation and mucosal integrity. In this preclinical study, aVNS was shown to improve several major pathophysiologies of FD: visceral hypersensitivity, delayed gastric emptying and hyperactive HPA axis in the rodent model of FD. These integrative effects of aVNS are unique to the proposed aVNS method and is of great clinical significance. In humans, aVNS can be replaced by the noninvasive taVNS due to the superficial innervation of the auricular vagus nerve. The taVNS is clinically feasible and there are already several clinical studies with the use of taVNS for FD [10], irritable bowel syndrome (IBS) [40] and adolescents with IBS [41]. A better understanding of the mechanisms of action involved with aVNS would lead to optimization of the taVNS methodology and promote taVNS as a non-pharmacological alternative therapy for FD.

### 4.7. Limitations

There were limitations in the present study. First, the EMG was recorded one week after the chronic intervention instead of two weeks after the intervention. This was due to the need for the placement of an intragastric balloon after the intervention and the need for the animal to recover after the placement of the gastric balloon. At the same time, in the preliminary experiment of our study, we observed that the damage of the gastric balloon was as high as 70% at the third week after the implantation of the balloon (two weeks after the intervention). Thus we recorded the EMG was in one week after the chronic intervention and one week after the surgery. The one-week delayed EMG test was also used in a previous study [42]. Secondly, the pattern of brain neuronal activation was not investigated in the present study. The activation of hypothalamus, NTS, locus coeruleus and raphe nucleus were reported with the electrical stimulation of the auricular vagus nerve in rodent models other than FD [43,44,45]. Thirdly, efferent and afferent vagal activity was not measured separately. We assessed the efferent t vagal activity via measuring gastric empty without the visceral sensitivity in the second experiment. Neurotracer techniques may be adopted in the future.

## 5. Conclusions

In the conclusion, aVNS with the special set of parameters exerts integrative effects on visceral hypersensitivity, delayed gastric emptying and depression-like behavior in IA-treated rats. These effects are closely related to the activation of anti-inflammation and the regulation of the humoral endocrine HPA axis (ACTH and corticosterone but not CFR or CRF1) anti-inflammatory mechanism mediated via the vago-vagal pathway. Therefore, there is potential for aVNS on both the primary outcomes of inflammatory diseases as well as comorbid psychiatric issues and quality of life measures, serving as a dual-pronged therapeutic. Data from this study suggest that targeting anti-inflammation may serve as a novel and worthy target for improving health outcomes associated with the FD.

## Figures and Tables

**Figure 1 brainsci-13-00253-f001:**
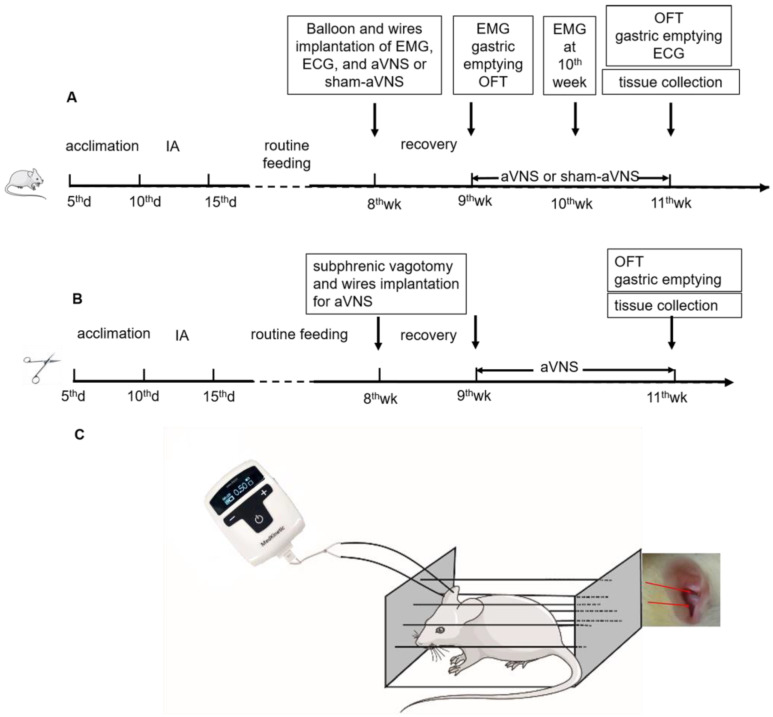
(**A**) Flowchart of experiment one. All ten-day-old SD rats except for control rats received iodoacetamide (IA) daily for 6 days and housed normally to grow until adulthood of 8 weeks old. Surgical implantations for ECG and EMG were performed at the age of 8 weeks and then recovered for 1 week. Physiological measurements were made at the 9th week and after the 11th week. Wire implantation for aVNS or sham-aVNS were performed at the age of 8 weeks and then recovered for 1 week at each group. The treatment (aVNS or sham-aVNS) was conducted for 2 weeks (9th week to 11th week). Only the EMG was performed after the 9th week and 10th week. (**B**) Flowchart of experiment two. All ten-day-old SD rats received the same IA treatment. Subphrenic vagotomy were performed at the age of 8 weeks and then recovered for 1 week. aVNS was conducted as experiment one. Physiological measurements were made after the 11th week. (**C**) aVNS schematic diagram.

**Figure 2 brainsci-13-00253-f002:**
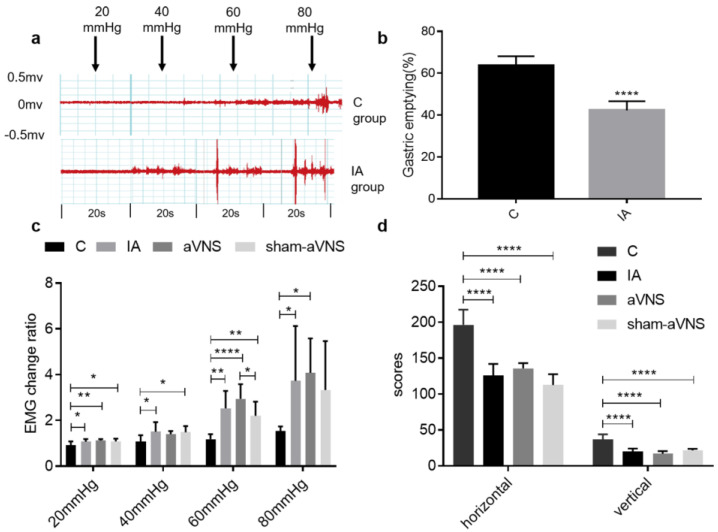
Assessment of rat model. (**a**) Different EMG waveforms changes response to gastric distension (GD) of 20, 40, 60 and 80 mmHg in control rats and IA-treated rats. (**b**) Gastric emptying results in control rats and IA-treated rats. The index was detected from 12 sacrificed rats after the 9th week (6 rats were randomly picked in control group and IA-treated group, respectively). The independent sample t-test was used. (**c**) EMG change ratio response to GD in all rats when rat model was induced. (**d**) The open field test (OFT) results when rat model was induced in all rats. All indexes were tested after the 9th week and before aVNS or sham-aVNS. One-way ANOVA was used, and the LSD method was used for post hoc tests in EMG change ratio and OFT. EMG: electromyography. * *p* < 0.05, ** *p* < 0.01, **** *p* < 0.0001. C: control group; IA: IA-treated group. *n* = 6 per group. Data were presented as means ± standard deviation.

**Figure 3 brainsci-13-00253-f003:**
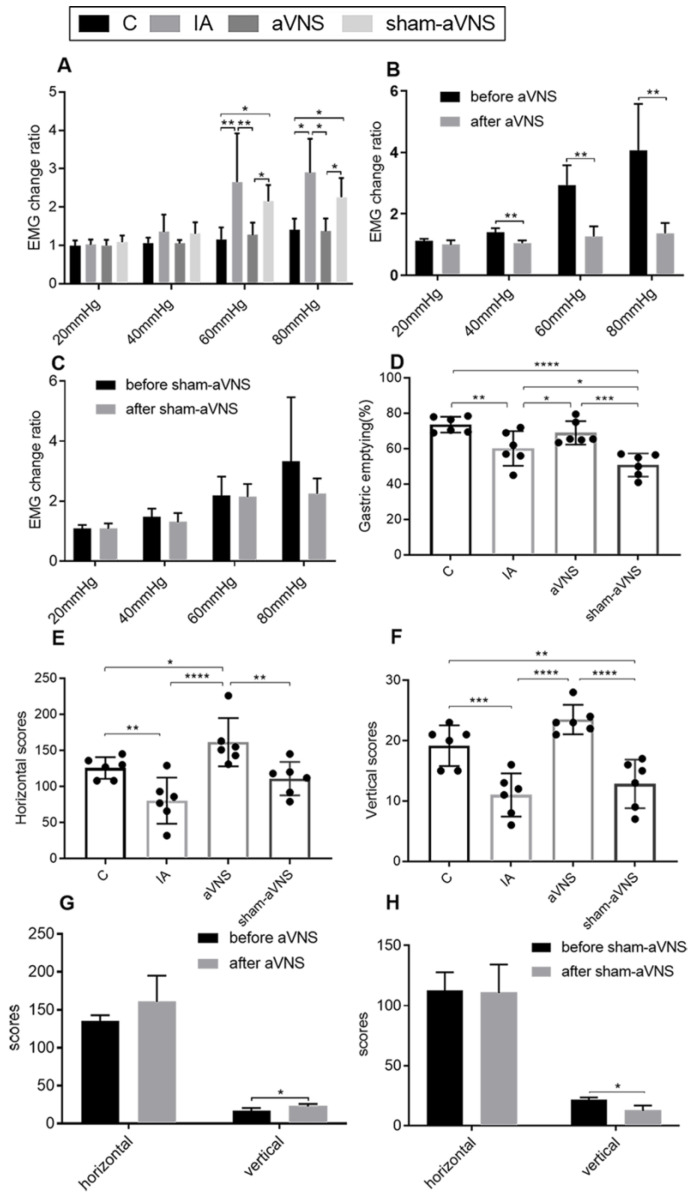
Effects of the aVNS. (**A**) EMG change ratio results after one-week intervention in all rats. (**B**) Comparison results of EMG change ratio before and after aVNS. (**C**) Comparison results of EMG change ratio before and after sham-aVNS. (**D**) Gastric emptying after two-week intervention in all rats. (**E**,**F**) OFT results after two-week intervention in all rats. (**G**) Comparison results of horizontal scores before and after aVNS. (**H**) Comparison results of vertical scores before and after sham-aVNS. A paired t-test was adopted for comparison in (**B**,**C**,**G**,**H**). One-way ANOVA was used, and the LSD method was used for post hoc tests in (**A**,**D**,**E**,**F**) (except for 80 mmHg in A Tamhane’s T2(M) method was used). EMG: electromyography. * *p* < 0.05, ** *p* < 0.01, *** *p* < 0.001, **** *p* < 0.0001. C: control group; IA: IA-treated group. *n* = 6 per group. Data were presented as means ± standard deviation.

**Figure 4 brainsci-13-00253-f004:**
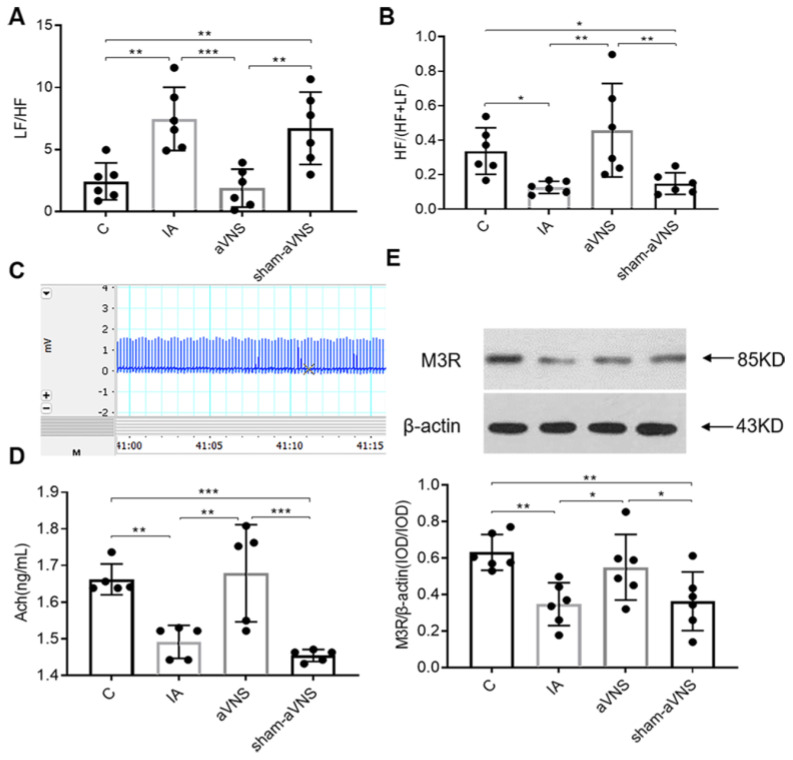
aVNS increased vagal activity, improved sympathovagal imbalance, promoted the level of Ach and its receptor of M3R in gastric tissue. (**A**) LF/HF results after two weeks of intervention in all rats. (**B**) HF/(HF + LF) results after two weeks of intervention in all rats. (**C**) ECG waveform in 15 s. (**D**) Acetylcholine concentration from gastric tissue in all results. (**E**) The expression of gastric M3R in all rats. LF: low frequency, represents sympathetic nerve activity. HF: high frequency represents vagal nerve activity. One-way ANOVA was used, and the LSD method was used for post hoc tests for all indexes. * *p* < 0.05, ** *p* < 0.01, *** *p* < 0.001. ECG: electrocardiograph. C: control group; IA: IA-treated group. *n* = 6 per group. Data were presented as means ± standard deviation.

**Figure 5 brainsci-13-00253-f005:**
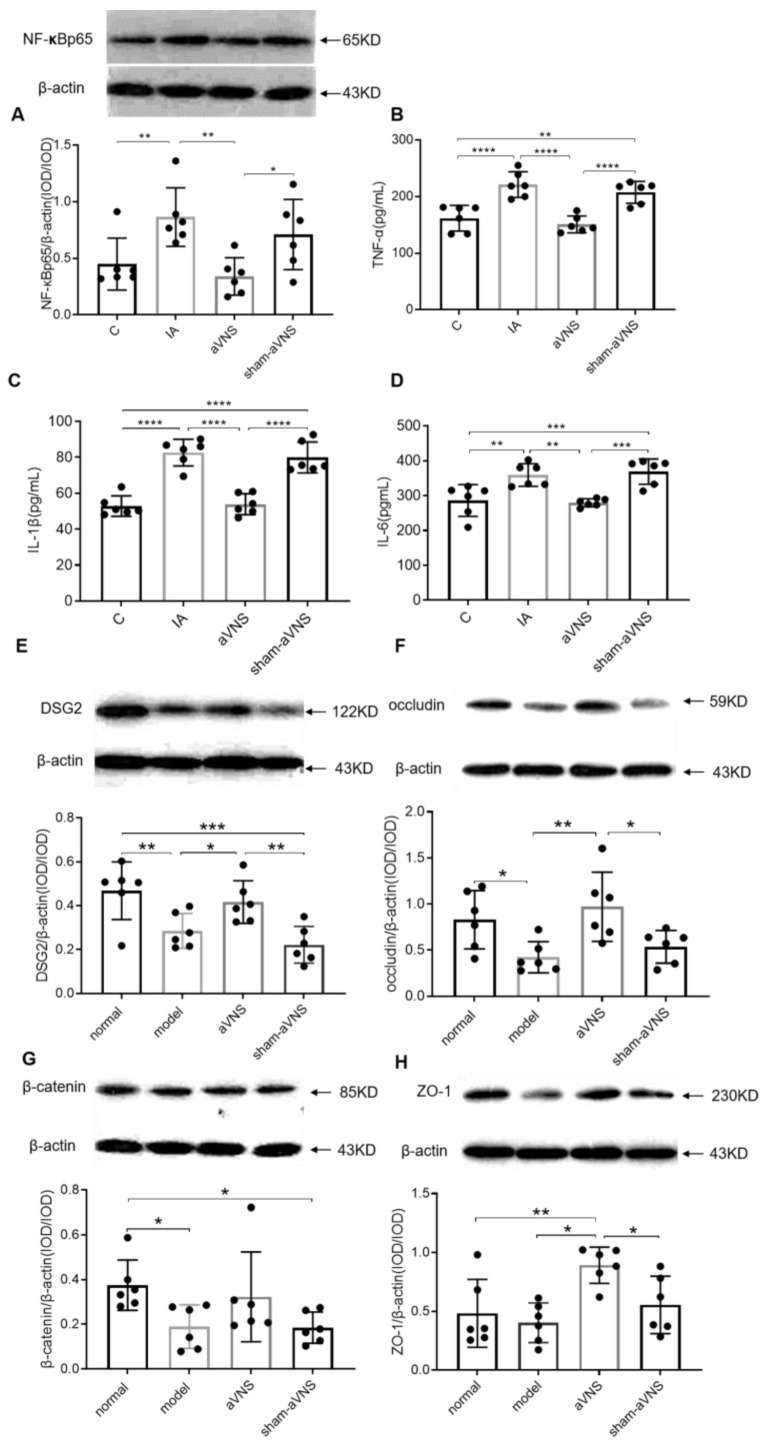
The anti-inflammatory and mucosal integrity mechanism of the intervention. (**A**) Level of duodenal NF-κBp65 after two weeks of intervention in all groups. (**B**–**D**) Release of serum TNF-α, IL-1β and IL-6 after two weeks of intervention in all groups. (**E**–**H**) Expressions of duodenal l proteins DSG2, occludin, β-catenin and ZO-1. One-way ANOVA was used, and the LSD method was used for post hoc tests for all. * *p* < 0.05, ** *p* < 0.01, *** *p* < 0.001, **** *p* < 0.0001. NF-κBp65: nuclear factor kappa Bp65.TNF-α: tumor necrosis factorα.IL-1β: interleukin 1β. IL-6: interleukin 6. DSG2: desmoglein2. ZO-1: zonula occluden. C: control group; IA: IA-treated group. *n* = 6 per group. Data were presented as means ± standard deviation.

**Figure 6 brainsci-13-00253-f006:**
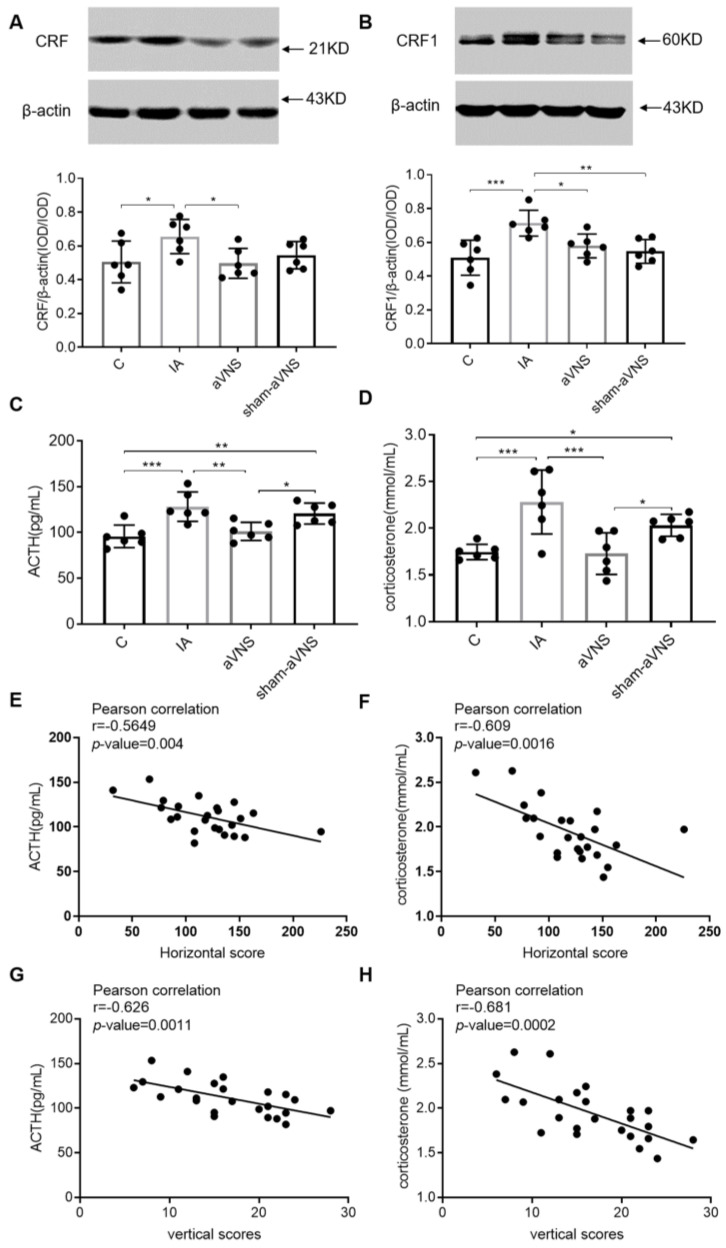
The hypothalamic–pituitary–adrenal axis mechanism of the intervention. (**A**) Expression of hypothalamus CRF in all rats. (**B**) Expression of amygdaloid nucleus CRF1 in all rats. (**C**,**D**) The release of serum ACTH and corticosterone in all rats. One-way ANOVA was used, and the LSD method was used for post hoc tests for all. (**E**–**H**): Pearson correlation analysis between HPA axis and OFT. * *p* < 0.05, ** *p* < 0.01, *** *p* < 0.001. ACTH: adrenocorticotropic hormone. CRF: corticotropin releasing factor. C: control group; IA: IA-treated group. *n* = 6 per group. Data were presented as means ± standard deviation.

**Figure 7 brainsci-13-00253-f007:**
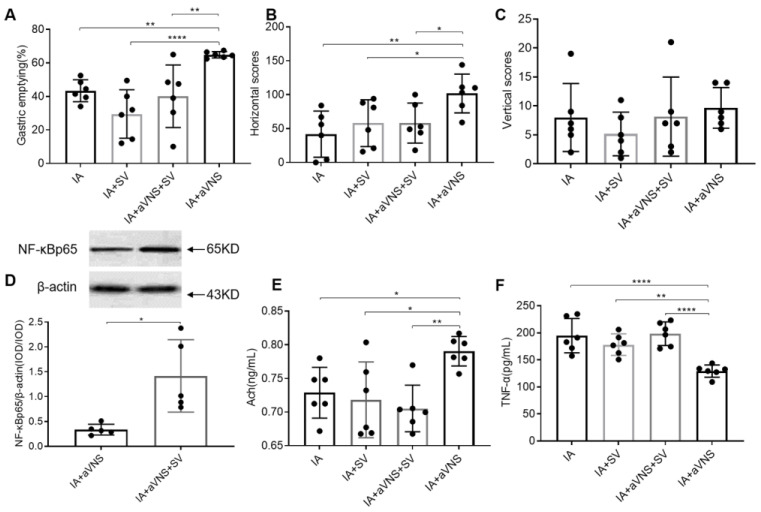
The effect and mechanism of vagotomy in IA-treated rats. (**A**–**C**) The gastric emptying and OFT results in all rats. (**D**) Expression of duodenal NF-κBp65 after two weeks of intervention. (**E**) The release of gastric Ach in all rats. (**F**) The serum level of inflammatory cytokines TNF-α in all rats. One-way ANOVA was used, and the LSD method was used for post hoc tests in (**A**–**C**,**E**,**F**). The independent sample t-test was used in (**D**). IA: IA-treated rats. IA + SV group: IA-treated rats received SV with no intervention. IA + aVNS group: IA-treated rats with 2 weeks of aVNS; IA + aVNS + SV group: IA-treated rats received SV with 2 weeks of aVNS. SV: subphrenic vagotomy. ** p* < 0.05, *** p* < 0.01, ***** p* < 0.0001. *n* = 6 per group. Data were presented as means ± standard deviation.

## Data Availability

Data are available upon reasonable request. All the relevant data generated and analyzed during the current study are available upon reasonable request to the corresponding author.

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
