# Peer review of "Auricular Vagus Nerve Stimulation Improves Visceral Hypersensitivity and Gastric Motility and Depression-like Behaviors via Vago-Vagal Pathway in a Rat Model of Functional Dyspepsia"

_brainsci, 2023, doi:10.3390/brainsci13020253_

Round 1
Reviewer 1 Report
Hou et al present a well-designed and well-conducted study showing how aVNS can alleviate the symptoms of functional dyspepsia by mitigating visceral hypersensitivity, aberrant gastric emptying, and depression-like behavior in a rat model of IA-induced functional dyspepsia. I hope the article would be helpful to the field to appreciate the role of the vagus nerve in regulating visceral physiology. I have some minor comments for the manuscript.
What happens to the food consumption in IA-treated animals with or without aVNS stimulation when the animals are given the food after the fasting?
Line 65: Please specify the method used for auricular stimulation in the cited literature.
Line 164: manufacturer names are missing for the ELISA kit.
Line 240 Full form of EMG should be used at the point it is first cited in the paper.
Figure 1B is too small and it is difficult to appreciate some of the changes visually in the graph.
Line 262 Please re-write Figure 4 legend title.
Please change the representative blots for 5F and 5H. The bands are contrasted too much.
Figure 7 fonts are too small to be legible
Line 466
"also used in previous" please check the grammar.
Author Response
Dear Reviewer,
Thank you very much for your comments. We have carefully read the comments and revised the manuscript point-by-point. The followings are our new answers.
- What happens to the food consumption in IA-treated animals with or without aVNS stimulation when the animals are given the food after the fasting?
Response: We adopted the gastric emptying to test the food consumption in under certain conditions. The result was from the euthanasia rat. This is the accepted method at present.
Reference: Zhou J, Li S, Wang Y, Lei Y, Foreman RD, Yin J, et al. Effects and mechanisms of auricular electroacupuncture on gastric hypersensitivity in a rodent model of functional dyspepsia. PLOS ONE 2017), 12:e174568.
Wang X, Yang B, Yin J, Wei W, Chen JDZ. Electroacupuncture via chronically implanted electrodes improves gastrointestinal motility by balancing sympathovagal activities in a rat model of constipation. Am J Physiol Gastrointest Liver Physiol. 2019 Jun 1;316(6):G797-G805.
- Line 65: Please specify the method used for auricular stimulation in the cited literature.
Response: We have specified the method in the manuscript.
- Line 164: manufacturer names are missing for the ELISA kit.
Response: We have added them in the manuscript.
- Line 240 Full form of EMG should be used at the point it is first cited in the paper.
Response: We have revised it in manuscript.
- Figure 1B is too small and it is difficult to appreciate some of the changes visually in the graph.
Response: We have revised it in manuscript.
- Line 262 Please re-write Figure 4 legend title.
Response: We have revised it in manuscript.
- Please change the representative blots for 5F and 5H. The bands are contrasted too much.
Response: We have revised it in manuscript.
- Figure 7 fonts are too small to be legible
Response: We have revised it in manuscript.
- Line 466 "also used in previous" please check the grammar.
Response: We have revised it in manuscript.

Reviewer 2 Report
Manuscript ID: brainsci-2042709
Title: Auricular vagus nerve stimulation improves visceral hypersensitivityand gastric motility via vago-vagal pathway in a rat model of functionaldyspepsia
Hou et al. present a very nice study exploring the therapeutic effects of auricular VNS in a rodent model of functional dyspepsia. The authors demonstrate that 1-2 weeks of this therapy is sufficient to normalize gastric emptying delay, visceral hypersensitivity, and depression-associated behaviors. They also show that aVNS led to decreased levels of serum inflammatory cytokines and duodenal mucosa NFkB levels, and increased levels of acetylcholine and tight junction proteins. Importantly, most of these effects were lost in rodents that underwent subdiaphragmatic vagotomy. The study is well conducted, however there are a number of concerns and clarifications needed before this manuscript can be accepted. In particular, the use of various protein measurements that the authors incorrectly attribute to physiologic functions/events.
Major concerns:
Figure 1: It is unclear if control rats underwent surgery, and which surgeries. Can this be clarified in the figure? For example, adding “control” next to “IA” during the neonatal period (if all rats then underwent surgery at week 8).
Figure 2:
- The authors report that EMG responses to gastric distension were increased in a pressure-dependent manner (Fig 2A, line 192), however the controls do not show this relationship (Fig 2A top recording): the EMG response is virtually the same at 20mHg, 40mmHg, and 60mmHg.
- Fig2B is very confusing and it is not clear why the authors include the pre-aVNS data. How is the IA group different from pre-aVNS and pre-aVNS sham (Line 195-7)?
Perhaps it would be best to simplify Fig2B by focusing on the control and IA comparison, and putting the week9 pre-aVNS rats in supplemental. The same comment applies to Fig 2D.
- Fig2C. The original FD model reported by Liu et al Gastro 2008 did not find any differences in gastric emptying between controls and IA (they only reported altered accommodation, but similar emptying rates). The authors need to address this discrepancy in their model, or include references (line 358) to back up this aspect of the model.
Figure 3:
- The authors observe an improvement after 1week of aVNS in tolerance to gastric distension. Did the authors observe similar results after 2 weeks of aVNS? If not measured, why not?
Figure 4:
- Can the authors confirm if the electrocardiogram data were recorded under similar circumstances for all groups, and what those circumstances were? i.e. morning, evening, fasting, fed, awake, anesthetized, etc.
Also, how long were the recordings used to calculate HRV measures? i.e. 2min, 5min 10min? This information can go in the methods and/or figure legend.
- Fig4D-E. As the majority of acetylcholine levels in the gut originates from enteric neurons, stomach tissue acetylcholine levels and receptor levels are not a valid surrogate for vagal activity. Unless the authors include references to indicate otherwise, (line 255-256), this interpretation should be rephrased. This also applies to discussion (lines 387) and elsewhere.
Figure 5:
- Why did the authors measure proteins in the duodenum rather than the stomach? There is no justification presented for this, or why this data is not presented.
- How was the duodenal mucosa dissected from the submucosa and muscularis? There are no controls to show these dissections were done uniformly.
- The authors measured total NFkB levels in the mucosa and describe this as a marker of inflammation. However, total levels of this protein are not a specific marker of mucosa inflammation: most cells express this protein. Its inflammatory function is demonstrated by post-translational modification and nuclear translocation, which the authors did not measure. If possible, the authors could present cytokine data (as they did in serum) to show that there was inflammation in the mucosa.
- Similarly to the point above, total tight junction protein levels are not surrogates for epithelial permeability. The function of these proteins is directly linked to their location on the basolateral plasma membrane, which was not studied. Permeability was also not studied. Thus, interpreting the results as measures of “gut permeability” is not accurate (Line 286) and results/discussion should be re-worded accordingly.
- Due to above, the title of Fig5 legend needs to be reworded.
Figure 6:
- Sham aVNS also reduced levels of brain CRF and CRF1. Similar significant results were observed for serum levels of ACTH and corticosterone. What is the difference between the IA group and the sham aVNS group?
- How were ACTH and corticosterone levels measured? RIA, ELISA? This information was not found in methods or figure legend.
Figure 7:
- The authors measured gastric emptying in vagotomized rats, and concluded that vagotomy abolished the normalization of stomach emptying seen in the IA aVNS group. This conclusion cannot be drawn, as it well established that subdiaphragmatic b/l vagotomy (SV) leads to pylorospasm and impaired gastric emptying in rodents. This is likely why the authors observed that SV led to worse gastric emptying compared to IA treatment alone, which could not be reversed by aVNS. In sum, gastric emptying (Fig7A) is not a reliable measure of aVNS in rats that have undergone SV.
- This led the reviewer to be concerned about the health of SV rats in the OFT in 7B-C (as it would be assumed that post SV, rats would have a gastric outlet obstruction and would not feed appropriately). This could confound the results. Were SV rats of comparable weight to the non-SV rats in figure 7B-C?
- The grouping of rats in Fig7 B, C, E, and F is confusing and difficult to follow as one reads the results text. Perhaps it would be clearer if the authors re-arrange the groups (AI -/+ aVNS, then AI+SV -/+ aVNS) and make it clear all 4 groups received AI.
- Fig7D is missing 2 control groups.
- The authors do not report on whether restoration of tight junction protein levels was abolished with SV. Can the authors comment or report this data?
Discussion: authors need to temper their conclusions regarding the relationship of the variables measured, which were not studied. For example, stating that visceral hypersensitivity was improved because of decreased gut permeability or inflammation (line 400). Or that vagotomy abolished prokinetic effects of aVNS (line 414).
Reference is made to vago-vagal (line ) and to efferent vagal activity, but the distinct branches of vagal activity were not measured.
The authors describe their anti-inflammatory findings as “CAP”, but this is not demonstrated in the study (blocking the inhibitory effect on serum cytokines using an acetylcholine receptor inhibitor for example). Thus, the authors can suggest that aVNS likely engaged CAP (as reported before), but not take this for granted when describing their results (example line 347).
Minor concerns:
Please explain sentence alluding to ref 3 (line 39-40)
Figure fonts are too small to read. Can these be made large font?
English grammar: see annotated PDF for examples.
Also, the word “mechanism” is used throughout the manuscript inappropriately, as no mechanistic studies were done. Therefore the variables studied cannot be attributed to cause and effect relationships.
The word “irritation” should be replaced by “stress” when describing the HPA axis (example line 297).
Please include reference for statement of HPA axis being an important “part” of the gut-brain axis (line 298). Technically, the HPA axis is not part of the gut-brain axis, which is conformed of the enteric nervous system, enteroendocrine cells, and the extrinsic nervous system of the gut (vagus and spinal cord).
Did the authors mean that the HPA axis is an important modulator of the gut-brain axis?
Fig2A and Fig 4C are lacking a time scale.
Fig2 Legend has letters B and C inverted.
Figure 5A:
- Authors measured protein levels by Western blot. Using the word “expression” in the text is confusing, as this suggests mRNA, which was not measured.
Figure 6:
- There is no introduction on CRF1, or why it was also measured (lines 297-300).
Figure 7E and discussion (line 387).
Acetylcholine “release” was not measured (this implies a stimulation with levels before and after in the same sample/animal). It is more appropriate to state acetylcholine levels in tissue were measured or similar.
Reference 23 (Gottfried-Blackmore et al) in line 381-382 is used incorrectly. The authors of that paper showed that nVNS led to improved abd pain symptoms, but they did not “hypothesize” that the improvement in abdominal pain following nVNS was due to parasympathetic reflex modulation. Please correct this citation accordingly.
- Discussion line 431-432: if vagal stimulation stimulates the HPA axis and increases ACTH, how is this consistent with the authors findings of aVNS decreasing ACTH?
Also, please cite primary literature showing that vagal stimulation stimulates the HPA axis. Ref 24 (Tracey et al) describes the CAP, which is another pathway not involving the HPA axis.
- Discussion line 395, authors did not study NFkB “pathway”, please correct.
- Discussion, Section 4.5: can the authors correlate the magnitude of HPA measures (CRF, ACTH, corticosterone) to the OFT behaviors for each rat? This may lend more credence that these two variables were connected, as they imply.
- Discussion line 433-435: This sentence is confusion and not clear. How do the authors associate changes in OFT with “efferent” vagal activity?
References are missing for the statements in discussion:
line 389 (CAP is one important function of vagus)
line 391 (mechanism of macrophage inhibition through vagal efferents).
Line 392 (vagus acetylcholine release in GI reticuloendothelial system)
Would include in “Limitations” that the distinct branches of vagal activity (efferent and afferent) were not measured.

Author Response
Dear Reviewer,
Thank you very much for your professional and rigorous comments. Through the deeper questions you suggested, we learned lots of new insights. Thank you for your kind attention in our study. We have carefully read the comments and revised the manuscript point-by-point. The followings are our new answers.
Major concerns:
- Figure 1: It is unclear if control rats underwent surgery, and which surgeries. Can this be clarified in the figure? For example, adding “control” next to “IA” during the neonatal period (if all rats then underwent surgery at week 8).
Response: All rats underwent surgeries including distension balloon implantation and electrodes implantation for EMG and ECG recording. We have revised the figure 1 legend to make it more clear in manuscript.
- Figure 2:
2.1 The authors report that EMG responses to gastric distension were increased in a pressure-dependent manner (Fig 2A, line 192), however the controls do not show this relationship (Fig 2A top recording): the EMG response is virtually the same at 20mHg, 40mmHg, and 60mmHg.
Response: Fig 2A: As the pressure increases, the EMG response to gastric distension (GD) becomes more obvious (Fig.2A), especially in IA-treated rats. There was almost no EMG response under 20 or 40 mmHg indeed in the control rat for its gastric sense was normal. We have revised the description in manuscript.
2.2 Fig2B is very confusing and it is not clear why the authors include the pre-aVNS data. How is the IA group different from pre-aVNS and pre-aVNS sham (Line 195-7)?
Perhaps it would be best to simplify Fig2B by focusing on the control and IA comparison, and putting the week9 pre-aVNS rats in supplemental. The same comment applies to Fig 2D.
Response: As you say, it was more clear if we only compared control and IA-treated rats, and the figure were concise. In addition to this, we wanted to describe that there were no difference among IA-treated, sham-aVNS and aVNS group before the 2-wk treatment. Thus we include the pre-aVNS and pre-sham aVNS dates.
2.3 Fig2C. The original FD model reported by Liu et al Gastro 2008 did not find any differences in gastric emptying between controls and IA (they only reported altered accommodation, but similar emptying rates). The authors need to address this discrepancy in their model, or include references (line 358) to back up this aspect of the model.
Response: In the original study, the gastric emptying method is different from ours. It adopted 3 hours foodintake to assess the gastric emptying. We used the 2g solid food in 10 minutes to test it, and the method is widely applied now.
- Figure 3: The authors observe an improvement after 1week of aVNS in tolerance to gastric distension. Did the authors observe similar results after 2 weeks of aVNS? If not measured, why not?
Response: This was due to the need for the placement of an intragastric balloon after the intervention and the need for the animal to recover after the placement of the gastric balloon. At the same time, in the preliminary experiment of our study, we observed that the damage of the gastric balloon was as high as 70% at the 3rd week after the implantation of the balloon (2 weeks after the intervention). Thus we recorded the EMG was in one week after the intervention to assess the effection and one week after the surgery to assess the model.
- Figure 4:
4.1 Can the authors confirm if the electrocardiogram data were recorded under similar circumstances for all groups, and what those circumstances were? i.e. morning, evening, fasting, fed, awake, anesthetized, etc. Also, how long were the recordings used to calculate HRV measures? i.e. 2min, 5min 10min? This information can go in the methods and/or figure legend.
Response: ECG was recorded for at least continuous 30 minutes in the day. The awake rat was kept in a restrainer in which the rat could only move legs slightly and could not turn around (same as Fig.1C). We have added it in the Appendix.
4.2 Fig4D-E. As the majority of acetylcholine levels in the gut originates from enteric neurons, stomach tissue acetylcholine levels and receptor levels are not a valid surrogate for vagal activity. Unless the authors include references to indicate otherwise, (line 255-256), this interpretation should be rephrased. This also applies to discussion (lines 387) and elsewhere.
Response: Thank you for your rigorous and professional advice. As you say, the gastric acetylcholine is just the neurotransmitter from enteric neurons or vagal nerve, it or its receptor level are not a valid surrogate for vagal activity. However, combination of acetylcholine and its receptor can promote gastric motility. In our study, only HRV from ECG represent the vagal activity. We have revised it in the manuscript.
- Figure 5:
5.1 Why did the authors measure proteins in the duodenum rather than the stomach? There is no justification presented for this, or why this data is not presented.
Response: Studies show that impaired intestinal barrier function and low-grade inflammation are athophysiological mechanism in FD patients. So we measure proteins in the duodenum rather than the stomach.
Reference: [1] Ford AC, Mahadeva S, Carbone MF, Lacy BE, Talley NJ. Functional dyspepsia. The Lancet 2020, 396: 1689-702.
[2] Vanheel, H., Vicario, M., Vanuytsel, T., Van Oudenhove, L., Martinez, C., Keita, Å. V., … Farré, R. (2013). Impaired duodenal mucosal integrity and low-grade inflammation in functional dyspepsia. Gut, 63(2), 262–271.
[3] Walker, M. M., & Talley, N. J. (2017). The Role of Duodenal Inflammation in Functional Dyspepsia. Journal of Clinical Gastroenterology, 51(1), 12–18.
5.2 How was the duodenal mucosa dissected from the submucosa and muscularis? There are no controls to show these dissections were done uniformly.
Response: We didn’t dissect the duodenal mucosa from the submucosa and muscularis.
Western Blotting was used to measure proteins.
5.3 The authors measured total NFkB levels in the mucosa and describe this as a marker of inflammation. However, total levels of this protein are not a specific marker of mucosa inflammation: most cells express this protein. Its inflammatory function is demonstrated by post-translational modification and nuclear translocation, which the authors did not measure. If possible, the authors could present cytokine data (as they did in serum) to show that there was inflammation in the mucosa.
Response: Thank you for your comments. We didn’t detect the inflammation in the mucosa. We have revised the description of NFkB and inflammation in the manuscript
5.4 Similarly to the point above, total tight junction protein levels are not surrogates for epithelial permeability. The function of these proteins is directly linked to their location on the basolateral plasma membrane, which was not studied. Permeability was also not studied. Thus, interpreting the results as measures of “gut permeability” is not accurate (Line 286) and results/discussion should be re-worded accordingly.
Response: We have revised the description in the manuscript.
5.5 Due to above, the title of Fig5 legend needs to be reworded.
Response: We have revised it in the manuscript.
- 6. Figure 6:
6.1 Sham aVNS also reduced levels of brain CRF and CRF1. Similar significant results were observed for serum levels of ACTH and corticosterone. What is the difference between the IA group and the sham aVNS group?
Response: Sham aVNS also reduced levels of brain CRF and CRF1. ACTH and corticosterone in serum were both increased in IA-treated rats and reduced with aVNS but not sham-aVNS. HPA axis is related with the irritation. The aVNS and Sham aVNS were both irritation for rats, maybe this is the underlying reason.
6.2 How were ACTH and corticosterone levels measured? RIA, ELISA? This information was not found in methods or figure legend.
Response: They were measured by ELISA. This information was in methods 2.6.1.
- Figure 7:
7.1 The authors measured gastric emptying in vagotomized rats, and concluded that vagotomy abolished the normalization of stomach emptying seen in the IA aVNS group. This conclusion cannot be drawn, as it well established that subdiaphragmatic b/l vagotomy (SV) leads to pylorospasm and impaired gastric emptying in rodents. This is likely why the authors observed that SV led to worse gastric emptying compared to IA treatment alone, which could not be reversed by aVNS. In sum, gastric emptying (Fig7A) is not a reliable measure of aVNS in rats that have undergone SV.
Response: Athough SV leads to pylorospasm and impaired gastric emptying in rodents, we didn’t find difference between (IA+SV) group and (IA+aVNS+SV) group. However, the IA+aVNS could improve the gastric emptying. The similar trend was found in other measurement.
7.2 This led the reviewer to be concerned about the health of SV rats in the OFT in 7B-C (as it would be assumed that post SV, rats would have a gastric outlet obstruction and would not feed appropriately). This could confound the results. Were SV rats of comparable weight to the non-SV rats in figure 7B-C?
Response: SV rats had a lower body weight than non-SV rats. But we didn’t compare their weight.
7.3 The grouping of rats in Fig7 B, C, E, and F is confusing and difficult to follow as one reads the results text. Perhaps it would be clearer if the authors re-arrange the groups (AI -/+ aVNS, then AI+SV -/+ aVNS) and make it clear all 4 groups received AI.
Response: The lengend was revised.
7.4 Fig7D is missing 2 control groups.
Response: We detected the NFkB protein levels in these two groups.
7.5 The authors do not report on whether restoration of tight junction protein levels was abolished with SV. Can the authors comment or report this data?
Response: We didn’t detect the tight junction protein levels in rats with SV.
7.6 Discussion: authors need to temper their conclusions regarding the relationship of the variables measured, which were not studied. For example, stating that visceral hypersensitivity was improved because of decreased gut permeability or inflammation (line 400). Or that vagotomy abolished prokinetic effects of aVNS (line 414).
Response: We have revised it in the manuscript.
7.7 Reference is made to vago-vagal (line ) and to efferent vagal activity, but the distinct branches of vagal activity were not measured.
Response: We didn’t measure the distinct branches of vagal activity. We have revised it in the manuscript.
7.8 The authors describe their anti-inflammatory findings as “CAP”, but this is not demonstrated in the study (blocking the inhibitory effect on serum cytokines using an acetylcholine receptor inhibitor for example). Thus, the authors can suggest that aVNS likely engaged CAP (as reported before), but not take this for granted when describing their results (example line 347).
Response: We replaced the description of CAP with the anti-inflammation. We have revised it in the manuscript.
Minor concerns:
- Please explain sentence alluding to ref 3 (line 39-40)
Response: The ref 3 showed some side effects of pharmacotherapy. We have revised 9.the sentence as “Adverse event profile of pharmacotherapy has recently been recognized in the guideline”.
- Figure fonts are too small to read. Can these be made large font?
Response: We have revised it in the manuscript.
- English grammar: see annotated PDF for examples.
Response:Thanks for your kind comments. We have revised it in the manuscript.
- Also, the word “mechanism” is used throughout the manuscript inappropriately, as no mechanistic studies were done. Therefore the variables studied cannot be attributed to cause and effect relationships.
Response: As you say, there were no cause and effect relationships. But the study had proved some mechanisms.
- The word “irritation” should be replaced by “stress” when describing the HPA axis (example line 297).
Response: We have revised it in the manuscript.
- Please include reference for statement of HPA axis being an important “part” of the gut-brain axis (line 298). Technically, the HPA axis is not part of the gut-brain axis, which is conformed of the enteric nervous system, enteroendocrine cells, and the extrinsic nervous system of the gut (vagus and spinal cord).
Did the authors mean that the HPA axis is an important modulator of the gut-brain axis?
Response: From the references, central signalling, including via corticotropin-releasing hormone and neurotransmitters, can alter peripheral gastrointestinal function. The HPA axis indeed wasn’t describe as an important “part” of the gut-brain axis. We have revised it in the manuscript.
References: Ford AC, Mahadeva S, Carbone MF, Lacy BE, Talley NJ. Functional dyspepsia. The Lancet 2020, 396: 1689-702.
- Fig2A and Fig 4C are lacking a time scale.
Response: We have revised it in the manuscript.
- Fig2 Legend has letters B and C inverted.
Response: We arranged the figure to satisify the uniform distribution.
- Figure 5A: Authors measured protein levels by Western blot. Using the word “expression” in the text is confusing, as this suggests mRNA, which was not measured.
Response: We have revised it in the manuscript.
- Figure 6: There is no introduction on CRF1, or why it was also measured (lines 297-300).
Response: The CRF1 in the amygdaloid nucleus was one important part of HPA axis, so we measured the CRF1.
- Figure 7E and discussion (line 387).
Acetylcholine “release” was not measured (this implies a stimulation with levels before and after in the same sample/animal). It is more appropriate to state acetylcholine levels in tissue were measured or similar.
Response: For the measure was detected when rats were killed. We didn’t measured the Acetylcholine in the same rat. We compare different group rats in the same time.
- Reference 23 (Gottfried-Blackmore et al) in line 381-382 is used incorrectly. The authors of that paper showed that nVNS led to improved abd pain symptoms, but they did not “hypothesize” that the improvement in abdominal pain following nVNS was due to parasympathetic reflex modulation. Please correct this citation accordingly.
Response: We replaced the reference as follows:
References: Zhu Y, Xu F, Lu D, Rong P, Cheng J, Li M, et al. Transcutaneous auricular vagal nerve stimulation improves functional dyspepsia by enhancing vagal efferent activity. Am J Physiol Gastrointest Liver Physiol 2021, 320:G700-11.
- Discussion line 431-432: if vagal stimulation stimulates the HPA axis and increases ACTH, how is this consistent with the authors findings of aVNS decreasing ACTH?
Also, please cite primary literature showing that vagal stimulation stimulates the HPA axis. Ref 24 (Tracey et al) describes the CAP, which is another pathway not involving the HPA axis.
Response: We red the reference carefully and considered that the s sentence” The vagal stimulation increases ACTH” is not valid, so we have deleted the sentene in the manuscript.
- Discussion line 395, authors did not study NFkB “pathway”, please correct.
Response: We have revised it in the manuscript.
- Discussion, Section 4.5: can the authors correlate the magnitude of HPA measures (CRF, ACTH, corticosterone) to the OFT behaviors for each rat? This may lend more credence that these two variables were connected, as they imply.
Response: We have revised it in the manuscript.
- Discussion line 433-435: This sentence is confusion and not clear. How do the authors associate changes in OFT with “efferent” vagal activity?
Response: The result suggested the vagal nerve not the efferent of vagal nerve was effective. We have deleted the “efferent” in the manscript.
- References are missing for the statements in discussion:
line 389 (CAP is one important function of vagus)
line 391 (mechanism of macrophage inhibition through vagal efferents).
Line 392 (vagus acetylcholine release in GI reticuloendothelial system)
Response: We have added the reference in the manscript.
- Would include in “Limitations” that the distinct branches of vagal activity (efferent and afferent) were not measured.
Response: We have added it in the limitations.

Reviewer 3 Report
In this study, Liwei Hou, et al. explored the effects of auricular Vagal Nerve Stimulation on gastric motility and visceral pain as well as depression-like behaviors and mechanisms involving cholinergic anti-inflammatory pathway (CAP) and the hypothalamic–pituitary–adrenal (HPA) axis in a rat model of Functional Dyspepsia (FD). The study findings are consistent with previous publications that vagal nerve stimulation can improve gastric motility and mental state of rats with FD, and the role of vagal gut-brain signaling in FD. The study is well performed. I would be interested in knowing if authors examined the tissue specimens to study the role of mast cells in FD rats.
Author Response
Dear Reviewers,
Thank you very much for your comments.
Reviewer 4 Report
Dear authors,
The objective of your work was to evaluate integrative effects and mechanisms of auricular vagus nerve stimulation (aVNS) in a rodent model of functional dyspepsia (FD). To achieve this goal, you resorted to an animal model of FD, and demonstrated aVNS effect in improving:
a) visceral hypersensitivity (probably due to activation of the cholinergic anti-inflammatory pathway and decreased mucosal permeability);
b) gastric motility (increasing efferent vagal activity) and
c) depression-like relief behaviors (probably by downregulation of HPA axis hyperactivity or the CRF brain signaling pathway).
Furthermore, the effects of aVNS were abolished by vagotomy, that:
a) blocked low-grade inflammation inhibition (demonstrating that the aVNS anti-inflammatory effect was mediated by the vagovagal reflex);
b) reversed aVNS prokinetic effect (demonstrating that vagus nerve integrity is important) and
c) hampered aVNS enhancing effects on open field test (OFT) horizontal movement, but not on vertical scores (suggesting that vagus nerve efferent projections may be one of the pathways to explain FD depression-like behaviours).
To obtain all these results several different methodologies were used, both in vitro and in vivo, enriching the presented work. However, the text must be revised so that English does not become a barrier to understanding the work. The title and abstract also deserved a revision that mirrors the work, and the "methods" section should also be improved.
1) Title: does not mention the (albeit more modest) behavioural improvements resulting from aVNS.
2) Abstract: should include the translation potential associated with this work...
3) 3) Experiments with animals:
a) why use only male rats?
b) when were the young rats separated from their mother and siblings?
c) line 98: replace the phrase "Rats in the control group did not receive any treatment" with "Rats in the control group did not receive any treatment"
d) line 12: replace the phrase "Ten-day-old SD rats were received iodoacetamide (IA) daily for 6 days and housed normally to grow until adulthood of 8-week-old. " with "Ten-day-old SD rats received iodoacetamide (IA) daily for 6 days and were housed normally to grow until adulthood of 8-week-old. "
e) Please indicate the success rate of inducing the FD model. What was the mortality rate associated with all protocols?
f) Why use such a long fasting (20h)? Have you tried shorter periods of time?
g) What was the euthanasia methos used?
4) Legend of figure 5 (E,F,G,H) - Identification of study groups different from the remaining figures.
5) When referring to muscarinic receptors subtype, authors should use “M3 muscarinic receptor” (the number that identifies the receptor is under scripted), according to IUPHAR’s “Concise Guide to PHARMACOLOGY 2021/22”
6) The horizontal and vertical scores obtained in the OFT are not explained in the “method” section, so the reader has to consult the “appendix” to understand what it is about...
7) Vagotomy in IA-treated rats abolished the anti-inflammation of aVNS, but only ACh, NF-κB and TNF-α were reported. Did you test cytokines (IL-6 and IL-1β, for example) in these conditions (SV rats)?
Thank you for presenting your work and I hope these suggestions are helpful.
I wish you luck and success.
Author Response
Dear Reviewers,
Thank you very much for your comments. We have carefully read the comments and revised the manuscript point-by-point. The followings are our new answers.
1) Title: does not mention the (albeit more modest) behavioural improvements resulting from aVNS.
Response: We thank the reviewer for your comment. For the improvements in the OFT behaviour is albeit more modest. And the vagotomy abolished the ameliorating effects of aVNS on horizontal motions but not vertical scores. These hinted that efferent projections of vagus nerve may be just one part of pathways in the ameliorating effect of aVNS for FD with depression-like behaviors. We wanted to emphasize its improved dynamics and hypersensitivity, so we didn't include it in the title.
2) Abstract: should include the translation potential associated with this work
Response: We have added it in the abstract.
3) Experiments with animals:
- a) why use only male rats?
- b) when were the young rats separated from their mother and siblings?
- c) line 98: replace the phrase "Rats in the control group did not receive any treatment" with "Rats in the control group did not receive any treatment"
- d) line 12: replace the phrase "Ten-day-old SD rats were received iodoacetamide (IA) daily for 6 days and housed normally to grow until adulthood of 8-week-old. " with "Ten-day-old SD rats received iodoacetamide (IA) daily for 6 days and were housed normally to grow until adulthood of 8-week-old. "
- e) Please indicate the success rate of inducing the FD model. What was the mortality rate associated with all protocols?
- f) Why use such a long fasting (20h)? Have you tried shorter periods of time?
- g) What was the euthanasia methos used?
Response: a) In the original model of FD, the study used the male rats to provide a potential model of chronic functional dyspepsia. In our study, we adopted the same method.
- b) In the three-week-old. We have added it in the manuscript.
- c) We have replaced the phrase.
- d) We have replaced the phrase.
- e) The model was with a successful modeling rate of 95%, we counted that in previous study. Reference: Hou L W,Rong P J,WeiW,Fang J L,WangD,Zhai W H,WangY,Wang J Y.Effect and mechanism study on transcutaneous auricular vagus nerve stimulation for functional dyspepsia model rats[J].World Journal of Acupuncture - Moxibustion,2020,30(1):49-56.
- f) We adopted the approved method which use such a long fasting (20h). We didn’t try the shorter time. Reference: Zhou J, Li S, Wang Y, Lei Y, Foreman RD, Yin J, et al. Effects and mechanisms of auricular electroacupuncture on gastric hypersensitivity in a rodent model of functional dyspepsia. PLOS ONE 2017), 12:e174568.
- g) At the end of the study, 90 min after feeding for assessing gastric emptying, all rats were anesthetized by general 2.5% isoflurane. Blood samples were collected from the ab-dominal aorta. Then the rat was died.
4) Legend of figure 5 (E,F,G,H) - Identification of study groups different from the remaining figures.
Response: We have revised it in manuscript.
5) When referring to muscarinic receptors subtype, authors should use “M3 muscarinic receptor” (the number that identifies the receptor is under scripted), according to IUPHAR’s “Concise Guide to PHARMACOLOGY 2021/22”
Response: We have revised it in the first time in manuscript.
6) The horizontal and vertical scores obtained in the OFT are not explained in the “method” section, so the reader has to consult the “appendix” to understand what it is about...
Response: Due to the word limit of the text, we had to put it in the appendix.
7) Vagotomy in IA-treated rats abolished the anti-inflammation of aVNS, but only ACh, NF-κB and TNF-α were reported. Did you test cytokines (IL-6 and IL-1β, for example) in these conditions (SV rats)?
Response: We didn’t tested other cytokines.

Round 2
Reviewer 2 Report
Manuscript ID: brainsci-2042709
Dear Hou et al. Thank you for revising this manuscript. You have a very nice study, but there are a few remaining concerns/clarifications remaining in the updated version of the manuscript.
Remaining clarifications:
Figure 1:
The authors have improved on the clarity of the figure, but it is still unclear when animals receive aVNS in relation to their surgeries and other interventions. For example, aVNS or sham are listed inside the box that includes surgeries at week 8, but then avNS appears again at week 9.5 through week 11. This is very confusing.
Adding a dark horizontal bar to indicate the exact days that animals were aVNS stimulated would be helpful.
This will also help to understand why some animals received 1 week of aVNS and others 2 weeks.
Finally, in their response, the authors say all rats had surgery, but the text indicates that a large number of rats did not undergo any treatment or surgery (line 106-111).
Perhaps it would be best to arrange Fig 1 in the same manner as the results are presented in the majority of the manuscript? That is, show a separate diagram for each of the 4 groups of mice: Control, IA, aVNS, and sham aVNS. This would assist the reader in interpreting the rest of the paper.
Right now, all 4 mouse groups are crammed into Fig1a. and it is very difficult to follow the text.
Figure 3:
It is still not clear when aVNS is being administered in relationship to the surgical procedures (relates to Fig1 as well).
The authors state that animals only received 1 week of aVNS because of the need to place an intragastric balloon after aVNS (“the intervention”). This does not make sense and is not the protocol that is explained in Fig 1.
One would expect that rats to be better healed at 2 weeks post balloon implantation than 1 week. Can this please be clarified? Perhaps it will be easier to understand if Fig.1 is revised.
Figure 5:
If the duodenal mucosa was not dissected from the muscularis, then the results and text should be modified to remove “mucosa” when describing levels in duodenum of NFkB, cytokines, etc. (Fig legend, line 288, 297, etc). Please remove "mucosa" from text.
Figure 6:
Can the authors please provide an explanation of why both aVNS and sham decreased levels of brain CRF and CRF1? This was asked but not answered.
If the reason is because both groups had stress of surgery and stimulation protocol, then the authors cannot conclude that aVNS decreases levels of these brain neurotransmitters (but rather it is the acute stress of the experiment). This limitation and explanation should be in the discussion section.
Figure 7:
The rationale and interpretation of results in vagotomized mice is still not justified or well clarified.
a) Given that subphrenic bilateral vagomoty severely impairs stomach emptying, how can the authors draw any conclusions of the effect of aVNS intervention on stomach emptying in these mice? The results provided in line 343 (Fig 7A) for non-vagotomized mice are the same as those from Figure 3D, but these do not address the limitation of measuring stomach emptying in vagotomized mice.
Figure 7D) If the authors measured and detected NFkB levels in the control groups, can this data please be presented in the figure?
Minor:
Time scale for Fig 4C is still missing.
Line 418 Discussion: NfKB was not “activated” by IA, its levels were increased. Can the authors please correct this distinction in through out the text?
English Spelling and Grammar checks.
Examples:
Line 271-73: “We detected gastric tissue levels of acetylcholine (Ach) and of its receptor M3R, as Ach and its receptor can promote gastric motility. Levels of Ach and M3R…”
Sentence 384-385 “…animal behaviors suggestive of depression.”
Line 438, “integrity” not “integrality”
Etc.
Question 23 was not answered: This reviewer is asking if there is a statistical association (or correlation) between the values in the OFT behavior results and the values in the ACTH, CRF, and corticosterone levels in the same rats.
Discussion, Section 4.5: can the authors correlate the magnitude of HPA measures (CRF, ACTH, corticosterone) to the OFT behaviors for each rat? This may lend more credence that these two variables were connected, as they imply.
Concern 26 (limitation that efferent and afferent vagal activity was not measured separately) was not addressed: The authors state that this information was added to the limitations section, but it was not.
Author Response
Dear Editors and Reviewers,
Thank you very much for your comments. We have carefully revised the manuscript. The followings are our answer.
Remaining clarifications:
Figure 1:
The authors have improved on the clarity of the figure, but it is still unclear when animals receive aVNS in relation to their surgeries and other interventions. For example, aVNS or sham are listed inside the box that includes surgeries at week 8, but then avNS appears again at week 9.5 through week 11. This is very confusing.
Adding a dark horizontal bar to indicate the exact days that animals were aVNS stimulated would be helpful.
This will also help to understand why some animals received 1 week of aVNS and others 2 weeks.
Response: We conducted the surgery of wires implantation of aVNS and sham at 8th week. The aVNS and sham were intervened from 9th week to 11th week. We have added a dark horizontal bar in the manuscript.
Finally, in their response, the authors say all rats had surgery, but the text indicates that a large number of rats did not undergo any treatment or surgery (line 106-111).
Response: Control rats were normal without any IA treatment and aVNS or sham, but they received the balloon and wires implantation of EMG and ECG. We revised in the experimental protocol and figure 1 legend.
Perhaps it would be best to arrange Fig 1 in the same manner as the results are presented in the majority of the manuscript? That is, show a separate diagram for each of the 4 groups of mice: Control, IA, aVNS, and sham aVNS. This would assist the reader in interpreting the rest of the paper.
Right now, all 4 mouse groups are crammed into Fig1a. and it is very difficult to follow the text.
Response: We have added the descriptions of four groups in the manuscript.
Figure 3:
It is still not clear when aVNS is being administered in relationship to the surgical procedures (relates to Fig1 as well).
The authors state that animals only received 1 week of aVNS because of the need to place an intragastric balloon after aVNS (“the intervention”). This does not make sense and is not the protocol that is explained in Fig 1.
One would expect that rats to be better healed at 2 weeks post balloon implantation than 1 week. Can this please be clarified? Perhaps it will be easier to understand if Fig.1 is revised.
Response: We revised Fig.1 and added descriptions of four groups in the manuscript to make it more clear. We conducted the surgery of balloon and wires implantation of aVNS and sham at 8th week before aVNS intervention. The aVNS and sham were stimulated from 9th week to 11th week. To assess the effect of the treatment, we detected EMG after one week of reatment at 10th week not 11th week for gastric balloon is easily damaged by stomach acid.
Figure 5:
If the duodenal mucosa was not dissected from the muscularis, then the results and text should be modified to remove “mucosa” when describing levels in duodenum of NFkB, cytokines, etc. (Fig legend, line 288, 297, etc). Please remove "mucosa" from text.
Response: We remove "mucosa" from text when describing levels in duodenum of NFkB and cytokines but retained mucosal integrity for the tight-junction protein reflects mucosal integrity.
Figure 6:
Can the authors please provide an explanation of why both aVNS and sham decreased levels of brain CRF and CRF1? This was asked but not answered.
If the reason is because both groups had stress of surgery and stimulation protocol, then the authors cannot conclude that aVNS decreases levels of these brain neurotransmitters (but rather it is the acute stress of the experiment). This limitation and explanation should be in the discussion section.
Response: Thank you very much for your comments. We have modified the conclusion and added the following statements in the Discussion: CRF and CRF1 were decreased in both groups of aVNS and sham-aVNS, suggesting that the increase (compared with the control rats) was probably attributed to stress effects of surgical and stimulation procedures.
In the conclusion, we have changed one sentence as follows: “These effects are closely related to the activation of anti-inflammation and the regulation of humoral endocrine HPA axis (ACTH and cortiscosterone but not CFR or CRF1) anti-inflammatory mechanism mediated via the vago-vagal pathway.”
Figure 7:
The rationale and interpretation of results in vagotomized mice is still not justified or well clarified.
- a) Given that subphrenic bilateral vagomoty severely impairs stomach emptying, how can the authors draw any conclusions of the effect of aVNS intervention on stomach emptying in these mice? The results provided in line 343 (Fig 7A) for non-vagotomized mice are the same as those from Figure 3D, but these do not address the limitation of measuring stomach emptying in vagotomized mice.
Response: Thanks a lot for your comment. We apologize for not making it clear. As you can see from Fig.7A, SV (vagotomy) delayed gastric emptying; however, there was no difference between the IA+SV group and the IA+SV+aVNS group, suggesting that at the presence of SV, aVNS was not capable of improving gastric emptying. To make it clear, the sentence in line 343 has been revised as follows: From Fig.7A, we can see that aVNS improved gastric emptying in IA-treated rats (P=0.007, IA+aVNS vs. IA). This improvement was however, not noted in the rats with SV (P>0.05, IA+aVNS+SV vs. IA+SV), suggesting the vagally mediated effect of aVNS.
Figure 7D) If the authors measured and detected NFkB levels in the control groups, can this data please be presented in the figure?
Response: We apologize for being unable to provide the control groups date for we just detected IA+aVNS group and IA+aVNS+SV group. This index were detected for supplement at the end. We considered underfunding problem and it has been detected in the first experiment, we just supplemented results of two groups.
Minor:
Time scale for Fig 4C is still missing.
Response: We have added time scale in the figure 4C.
Line 418 Discussion: NfKB was not “activated” by IA, its levels were increased. Can the authors please correct this distinction in through out the text?
Response: We have revised them in the manuscript.
English Spelling and Grammar checks.
Examples:
Line 271-73: “We detected gastric tissue levels of acetylcholine (Ach) and of its receptor M3R, as Ach and its receptor can promote gastric motility. Levels of Ach and M3R…”
Sentence 384-385 “…animal behaviors suggestive of depression.”
Line 438, “integrity” not “integrality”
Etc.
Response: We have revised them in whole text.
Question 23 was not answered: This reviewer is asking if there is a statistical association (or correlation) between the values in the OFT behavior results and the values in the ACTH, CRF, and corticosterone levels in the same rats.
Discussion, Section 4.5: can the authors correlate the magnitude of HPA measures (CRF, ACTH, corticosterone) to the OFT behaviors for each rat? This may lend more credence that these two variables were connected, as they imply.
Response: For CRF and CRF1 were decreased in both groups of aVNS and sham-aVNS, we consider effects are closely related to the regulation of humoral endocrine HPA axis (ACTH and cortiscosterone but not CFR or CRF1). We correlate HPA measures (ACTH, corticosterone) to the OFT behaviors for each rat in figure 6. The results showed the negative linear correlation between HPA measures and OFT behaviors.
Concern 26 (limitation that efferent and afferent vagal activity was not measured separately) was not addressed: The authors state that this information was added to the limitations section, but it was not.
Response: Thanks for your comment. We have added it in the limitation.
